# G Protein-Coupled Estrogen Receptor (GPER) and ERs Are Modulated in the Testis–Epididymal Complex in the Normal and Cryptorchid Dog

**DOI:** 10.3390/vetsci11010021

**Published:** 2024-01-05

**Authors:** Giovanna Liguori, Simona Tafuri, Alessandra Pelagalli, Sabrina Ali’, Marco Russo, Nicola Mirabella, Caterina Squillacioti

**Affiliations:** 1Department of Veterinary Medicine and Animal Production, University of Napoli Federico II, 80137 Naples, Italy; giovanna.liguori@unina.it (G.L.); simona.tafuri@unina.it (S.T.); ali@unina.it (S.A.); marco.russo@unina.it (M.R.); nicola.mirabella@unina.it (N.M.); caterina.squillacioti@unina.it (C.S.); 2Department of Prevention, ASL FG, Piazza Pavoncelli 11, 71121 Foggia, Italy; 3Department of Advanced Biomedical Sciences, University of Napoli Federico II, 80137 Naples, Italy; 4Institute of Biostructures and Bioimages, National Research Council, Via De Amicis 95, 80131 Naples, Italy

**Keywords:** estrogen receptors, G protein-coupled estrogen receptor, dog male genital tract, cryptorchidism, oxidative stress

## Abstract

**Simple Summary:**

The testicular function as well as the maintenance and control of spermatogenesis are regulated by a delicate balance between androgens and estrogens. Cryptorchidism, largely diffused in canine species, is a congenital abnormality of the genitourinary tract, due to the failure to descend by one or both testes into the scrotal sac. In consideration of the possible role of estrogenic molecules affecting testicular descent, the aim of this study was to determine the distribution and expression (proteins and relative mRNAs levels) of two nuclear estrogen receptors (ERs), ER-alpha and ER-beta and a trans-membrane G protein-coupled estrogen receptor (GPER), in the testis–epididymal complex of the dog. In addition, in these tissues the expression level of two proteins as SOD1 and Nrf2 normally associated with oxidative stress was investigated to evaluate possible relation with ERs. Collectively, the results obtained by using Immunohistochemistry, Western blot and qRT-PCR showed changes in the distribution and expression of the GPER and ERs between the normal and cryptorchid dog. In detail, an upregulation of GPER and ER-alpha and a downregulation of ER-beta in the canine cryptorchid reproductive tract was observed in association with a modulation of SOD1 and Nrf2 expression.

**Abstract:**

There is growing evidence by the literature that the unbalance between androgens and estrogens is a relevant condition associated with a common canine reproductive disorder known as cryptorchidism. The role of estrogens in regulating testicular cell function and reproductive events is supposedly due to the wide expression of two nuclear estrogen receptors (ERs), ER-alpha and ER-beta and a trans-membrane G protein-coupled estrogen receptor (GPER) in the testis. In this study, immunohistochemistry, Western blotting and qRT-PCR were used to assess the distribution and expression of GPER in the testis–epididymal complex in the normal and cryptorchid dog. ER-alpha and ER-beta were also evaluated to better characterize the relative abundances of all three receptors. In addition, in these tissues, the expression level of two proteins as SOD1 and Nrf2 normally associated with oxidative stress was investigated to evaluate a possible relationship with ERs. Our data revealed changes in the distribution and expression of the GPER between the normal and cryptorchid dog. In particular, dogs affected by cryptorchidism showed an upregulation of GPER at level of the examined reproductive tract. Also considering the obtained result of a modulation of SOD1 and Nrf2 expression, we could hypothesize the involvement of GPER in the cryptorchid condition. Further studies are, however, necessary to characterize the role of GPER and its specific signaling mechanisms.

## 1. Introduction

Cryptorchidism is when one or both testicles fail to descend from the abdominal cavity into the scrotum and is a condition that largely affects the canine population [1,2]. This condition is common also in other species (stallions, boars, and humans) [3,4,5,6]. The etiology of this disorder is not completely known, but several studies suggest that genetic predisposition and environmental risk factors could contribute to the onset of disease [2]. Cryptorchidism is associated with morphological alterations of tubular and interstitial compartments of the testis and epididymis [7,8] due to the fact that the undescended testis exposed to a body temperature higher than the scrotal temperature may impair sperm production [9,10]. In addition, this condition may be associated with endocrine disorders characterized by both decreased testicular concentrations and increased estradiol concentrations in the peripheral blood circulation, as reported in dogs [11]. Interestingly, cryptorchidism predisposes individuals to testicular tumors, and estrogen levels in dogs with tumors are higher than in cryptorchid dogs [11,12]. 

Considering that estrogens are steroid hormones regulating many reproductive functions as well as a range of other physiological functions in both sexes, their role is pivotal in maintaining the morpho-functional integrity of the male reproductive tract [13,14]. Furthermore, estrogens are generated mainly by the action of aromatase, which converts testosterone into estradiol and androstenedione into estrone, respectively. This process is particularly evident for in vivo cryptorchidism in horses and in mice at the level of the testis [15], epididymal duct, and the prostate. In our previous study, aromatase expression at level of the compartments (interstitial and tubular) of the normal and cryptorchid testis was demonstrated. In particular, we demonstrated an increase in aromatase activity in the canine retained male gonad [16]. Traditionally, two nuclear estrogen receptors (ERs), ER-alpha and ER-beta, considered ligand-activated transcription factors, are considered the main players in the activity of 17-beta-estradiol. However, 17-beta-estradiol also helps the signaling events of pathways involving transmembrane ERs, such as G-protein-coupled ER 1 (GPER; formerly known as GPR30). GPER is a widely conserved 7-transmembrane-domain protein and is structurally distinct from classic ERs, showing a binding domain at level of the plasma membrane and endoplasmic reticulum [17,18]. Functionally, it mediates rapid non-genomic estrogen signaling [19,20], and it plays different roles in both normal and cancer cells [21], including testis physiology regulation [22].

GPER expression was investigated in the testis of the horse [23], rat [24,25,26] and human [27], while in the epididymis, its expression was reported in pigs [28], humans [29] and rats [30]. On the contrary, in dogs, limited studies have shown its presence in testes with and without tumors [31], and no data are available for dog epididymis. On the other hand, the pattern of classic estrogen receptors’ protein abundance was studied in the male reproductive tracts of some animal species including dogs [32,33,34], humans [35], and rodents [26,36], evidencing some differences in their distribution. In our opinion, knowledge is limited regarding both the tissue-specific distribution and coexistence of these three estrogen receptors in normal and cryptorchid conditions in dogs. 

The aim of the study was to determine the level of ERs and GPER in mRNAs, their protein expression, and their distribution in the testis–epididymis complex, and also to determine any difference between normal and cryptorchid dogs. In addition, we address the question as to whether GPER expression is related to the oxidative status of this tissue by investigating the levels of two proteins, SOD1 and Nrf2. Estrogens, through their receptors, may be implicated in the regulation of biological processes (cellular proliferation, metabolic activity, and reproduction). For this reason, estrogen–stress interactions are considered crucial to our understanding of the antioxidant actions of tissues that perform reproductive functions [37]. It is well known that oxidative stress, determined by an imbalance between pro- and anti-oxidants, can alter the signaling pathways controlling cellular function [38]. Regarding physiological conditions, the ROS level is balanced when excessive free radicals are scavenged by antioxidant enzymes such as superoxide dismutase (SOD) [39]. As recently reported, ROS or other free radicals can disturb the mechanisms controlling cell functions through an alteration of transcription factors and of the redox status signaling pathway. In this condition, Nrf2 acts by regulating the expression of various antioxidant enzymes to counteract the oxidative stress condition and thus maintain cellular redox homeostasis [40]. Of note, the possible relationship between the enzyme SOD1 and the Nrf2 antioxidant system has been demonstrated by Kirby et al. [41] showing that the presence of mutSOD1 in a mouse motor neuron-like cell line resulted in reduced Nrf2 mRNA expression and downregulation of Nrf2 target genes. These findings may contribute the possible implications of these receptors in the functions of this complex both in the normal and cryptorchid condition. 

## 2. Materials and Methods

### 2.1. Animals 

The current research was performed on testis–epididymal complex obtained from a total of n. 20 adult male mixed-breed dogs divided into two groups: control (*n* = 10) and cryptorchid (*n* = 10). In detail, the control group was identified by mature healthy dogs (average weight 19.8 ± 2.7 kg, average age = 4.8 ± 1.91 years), while the cryptorchid group was represented by dogs with unilateral cryptorchidism (testis retained in the abdomen/the inguinal canal) (average weight 18.0 ± 2.0 kg, average age = 4.2 ± 1.64 years). The aforementioned subjects had undergone orchiectomy at the Veterinary Clinic of the University Federico II, Naples, Italy. To enroll dogs in the study, the written consent of their owners was obtained. The experimental procedures were compliant with and approved by the Ethical Committee for the Care and Use of Animals of the University of Naples Federico II, Department of Veterinary Medicine and Animal Production, Naples (no. 0-050-377). Dogs involved in this research were not previously used in any clinical trials or treatments. 

### 2.2. Tissue Collection 

The testis–epididymal complex was surgically removed from each dog (from control and cryptorchid groups). In cryptorchid dogs, the scrotal testis was normal in size and morphology, while the ectopic testis was always small in size, but without macroscopic lesions resulting from neoplastic processes. Then, the epididymis was divided into three portions: the (1) caput, (2) corpus, and (3) cauda. After separation from the testis, the entire caput and cauda epididymal portions were separated through careful macroscopic observation of the organ. For the immunohistochemical analysis, all the collected tissues (from *n* = 5 dogs for each group) were firstly fixed by immersion in Bouin’s fluid for 12–24 h at room temperature, processed for paraffin embedding in a vacuum, and then cut at a thickness of 6–7 microns. For Western blotting and real time RT-PCR, all samples (from the remaining *n* = 5 dogs for each group) were collected and directly frozen in dry ice and stored at −80 °C until use.

### 2.3. Immunohistochemistry

The tissue sections (7 μm thick), after deparaffination and rehydration, were incubated in 3% H_2_O_2_ for 20 min at room temperature (RT). After blocking the non-specific sites in 1.5% normal goat serum (NGS) diluted with phosphate-buffered saline (PBS) pH 7.2 (NGS cod S1000, Vector Laboratories, Burlingame, CA, USA), sections were incubated with rabbit polyclonal anti-GPR30 antibody (dil. 1:500, cod. PA5-28647, Invitrogen, Waltham, MA, USA), mouse monoclonal anti-ESR1 (dil. 1:200, cod. TA807239, OriGene Technologies Inc., Rockville, MD, USA), and rabbit polyclonal anti-ER-beta (dil. 1:500, cod. PA1-311, Invitrogen, Waltham, MA, USA) overnight at 4 °C.

After washing in PBS, the sections were incubated with HRP-conjugated secondary antibody (dil. 1:4, cod. UNIHRP-050 ImmunoReagents, Raleigh, NC, USA) for 30 min at RT. The immunoreactions were developed with 3,3′-diaminobenzidine tetrahydrochloride (DAB) (SK-4100, Vector Laboratories, Burlingame, CA, USA). In addition, the sections were counterstained by using hematoxylin. The negative controls included omission of primary antibody (Appendix A).

The immunoreactions were observed by two different blind observers who then captured images and analyzed them using a Leica DM 6B light microscope and SFC7000T digital camera.

### 2.4. Western Blot

Homogenization of frozen tissues was performed in ice-cold RIPA buffer with 1× protease cocktail inhibitors, and the suspension of homogenized tissues was centrifuged at 14,000 rpm for 30 min at 4 °C for protein extraction. The protein concentration of samples was measured using a Bradford assay (Bio-Rad Laboratories Inc., Hercules, CA, USA). Equal protein amounts (30 μg) dissolved in loading buffer were separated on a 4–20% Mini-PROTEAN, TGX Stain-Free Precast Electrophoresis Gel (Bio-Rad Laboratories, Inc., Hercules, CA, USA). After electrophoresis, proteins were electro-transferred to a nitrocellulose membrane for assessment of protein transfer using a ChemiDoc Molecular Imager (Bio-Rad Laboratories, Inc., Hercules, CA, USA). After being saturated in 5% non-fat milk and diluted in TBS-T (1.5 M NaCl, 200 mM Tris-HCl, and 0.1% Tween-20, pH 7.2) for 1 h at RT, the blots were incubated with different primary antibodies which were diluted 1:500 in 2.5% non-fat milk-TBS-T (rabbit polyclonal anti-GPR30 antibody, cod. PA5-28647, Invitrogen, Waltham, MA, USA; mouse monoclonal anti-ESR1, cod. TA807239, OriGene Technologies, Inc., Rockville, MD, USA; rabbit polyclonal anti ER-beta, cod. PA1-311, Invitrogen, Waltham, MA, USA; rabbit polyclonal anti-SOD1, orb 375349, Biorbyt Ltd. Cambridge, United Kingdom; mouse monoclonal anti-Nrf2-(A10), sc. 365949, Santa Cruz Biotechnology, Inc., Dallas, TX, USA) overnight at 4 °C.

The following day, after washing in TBS-T, the blots were incubated with secondary antibody (goat anti-rabbit or anti-mouse) conjugated with horseradish peroxidase (HRP) (ImmunoReagents, Raleigh, NC, USA), diluted 1:1000 in 2.5% non-fat milk-TBS-T) for 1 h at RT. After the last three washes with TBS-T, protein detection was performed using ECL (Bio-Rad Laboratories, Inc., Hercules, CA, USA), and an image was captured by the ChemiDoc Molecular Imager (Bio-Rad Laboratories, Inc., Hercules, CA, USA). The standard molecular weight marker used was Precision Plus ProteinTM All Blue Prestained Protein Standards (10–250 KdA, cod. #1610373, Bio-Rad Laboratories, Inc., Hercules, CA, USA). Densitometric analysis was performed using the Image Lab software version 6 (Bio-Rad, Hercules, CA, USA) with normalization to the total protein concentration for each lane. 

The results are expressed as the relative intensity to that of the normal tract for testis and each epididymal tract. In addition, for densitometric analysis in the different epididymal segments, the results are expressed as relative intensity to caput epididymis.

### 2.5. RNA Isolation, cDNA Synthesis, and Real-Time RT-PCR

Total RNA extraction was performed from the tissue samples in ice-cold Trizol reagent (Invitrogen, Waltham, MA, USA) following the manufacturer’s protocol and an Ultra-Turrax homogeniser. Total RNA (1 μg) was retrotranscribed using a high-capacity cDNA reverse transcription kit (Applied Biosystems, Carlsbad, CA, USA) according to the manufacturer’s instructions, using random hexamers as primers. Specific primers that amplify specific regions of the GPER, ER-alpha and ER-beta genes were designed using the published GenBank gene sequences and Primer express Software v.3.0.1. The GenBank accession number and sequences of the primers are listed in Table 1. The real-time PCR reactions contained 1 μL of cDNA (40 ng/well) and 24 μL of SYBR Green Master Mix (Applied Biosystems, Carlsbad, CA, USA) containing specific primers.

The conditions used for PCR were the follows: 50 °C for 2 min, 94 °C for 10 min, followed by 40 cycles at 94 °C for 15 s and 60 °C for 1 min. The reference gene GAPDH was amplified in separate wells using the same conditions. Real-time detection was performed using an ABIPRISM 7300 Sequence Detection System (Applied Biosystem, Foster City, CA, USA), and data from the SYBR Green PCR amplicons were assessed using ABI 7300 System SDS Software. The relative quantification 2^−ΔΔCt^ method was used for determination of the relative mRNA transcript abundance, as described previously [42].

### 2.6. Statistical Analysis

Data expressed as the mean ± standard error (SE) were used for densitometric and quantitative RT-PCR analyses. To determine differences in protein or mRNA transcript abundances between different segments of the epididymis in dogs with and without cryptorchidism, a one-way analysis of variance (ANOVA) and Tukey’s honestly significant difference (HDS) test for the independent samples were used. The differences between values for variables in tissues of dogs with and without cryptorchidism were determined using an unpaired student’s *t*-test. Differences were considered to be statistically significant if *p* was <0.05 when data were analyzed for all experiments. Calculations for statistical analyses were performed using SPSS software program version 25.0 (IBM, Armonk, NY, USA).

## 3. Results

### 3.1. Immunohistochemical Evaluation of GPER, ER-Alpha and ER-Beta in the Normal and Cryptorchid Testis of Dogs

In the normal testis, GPER immunoreactivity (IR) was localized both in germinal and interstitial compartments (Figure 1). In particular, GPER-IR was found in pachytene primary spermatocytes (Figure 1a; arrows) and in few oval spermatids (Figure 1a; line arrows). In pachytene spermatocytes, GPER-IR assumed the aspect of a roundish granule which was localized in a close perinuclear position. Along the spermatid maturation, the positive material changed in shape and localization following the morphological transformation of the cell which, as known, is round in the young elements and becomes progressively more oval and elongated in the older ones. Sertoli cells and round spermatids were negative.

ER-alpha was distributed in germ cells and in the interstitial Leydig cells. In the germinal compartment of the testis, ER-alpha-IR was found in numerous round spermatids (Figure 1b; R).

ER-beta IR was distributed in Sertoli cells (Figure 1c; arrowheads), where their cytoplasm enveloped a group of elongated or mature spermatids (Figure 1c; double arrows, insert). Moreover, ER-beta-IR was detected in the interstitial Leydig cells (Figure 1b; Lc) and in the tubular compartment. In fact, young spermatids showed a semilunar-shaped positive structure closely adherent to the nuclear membrane (Figure 1b). Elongated spermatids showed a granule in their tail, which was constantly turned towards the tubular lumen (Figure 1c).

In the cryptorchid male gonad, GPER-IR was distributed in both the interstitial Leydig cells (Figure 1d; Lc) and Sertoli cells (Figure 1d; arrowheads) while the pre-meiotic cells were negative. In addition, ER-alpha- and ER-beta-IR were distributed with a pattern similar to GPER (Figure 1e,f). In particular, Sertoli cells’ ER-beta-IR contents were characterized by intensely stained granules that completely filled the cytoplasm of the cells (Figure 1f; arrowheads).

The localization and semi-quantitative evaluation of GPER, ER-alpha, and ER-beta in the male gonad of normal and cryptorchid dogs are summarized in Table 2.

### 3.2. Immunohistochemical Evaluation of GPER, ER-Alpha, and ER-Beta in the Normal and Cryptorchid Epididymis of Dogs

In the normal epididymis, GPER-IR was found in the epithelial cells of all three segments (Figure 2) and in the ciliated cells of the efferent ductules (Figure 2; insert). In particular, GPER-IR was found in the apical portion of the principal epithelial cells and in the stereocilia-covered surface of these epithelial cells in the caput and cauda (Figure 2a,m; arrows) while the peritubular muscular cells of the corpus and cauda were also immunoreactive to GPER (Figure 2a,g,m; asterisk). ER-alpha- and ER-beta-IR were found in epithelial cells of the different segments. In detail, ER-alpha-IR was found in the epithelial ciliated cells (nuclear signal) and in some nonciliated type B cells containing vacuoles of the efferent ductules (nuclear signal) (Figure 2b; double arrows) and in the apical portion of the principal cells of the all segments (Figure 2b,h,n; arrows). Differently, only the basal cells of all segments were positive for ER-beta (Figure 2c,i,o; arrowheads). In addition, some positive cells were entirely positive from their basal zone to the apical zone (Figure 2c,i,o; inserts).

In the cryptorchid epididymis, GPER-IR was distributed in the epithelial cells of all epididymal segments, with a distribution pattern like that of the normal epididymis (Figure 2d,j,p, Table 3). In particular, GPER-IR was also distributed in the peritubular muscular cells in the cryptorchid corpus (Figure 2j, asterisks) and cauda (Figure 2p, asterisks), and the number of positive structures was higher than in normal segments (Figure 2). ER-alpha- and ER-beta-IR were found in epithelial cells of the different segments. The principal cells of the caput (Figure 2e, arrows) and corpus (Figure 2k, arrows) in their apical portion were immunoreactive to ER-alpha-IR, while in the cauda, the immunoreactivity was located in the nuclei of principal epithelial cells (Figure 2q). ER-beta-IR was found both in the basal cells and apical portions of the epithelial principal cells of the caput and corpus (Figure 2f,l), while in the cauda, only the apical portion of the principal cells showed a positive result (Figure 2r). Some positive epithelial cells completely filled with condensed granules, which defined the complete profile of the cells, were found in the corpus (Figure 2l-insert). The peritubular muscular cells were negative. Table 3 shows the immunostaining pattern of GPER, ER-alpha, and ER-beta in the epididymal segments of normal and cryptorchid dogs.

### 3.3. Protein Expression Levels of GPER, ER-Alpha and ER-Beta in the Normal and Cryptorchid Testis–Epididymal Complex: Western Blot and Densitometric Analysis

As shown in Figure 3, GPER, ER-alpha, and ER-beta were expressed in all examined tissues albeit with a different regulation between normal and cryptorchid dogs. In particular, GPER and ER-alpha expression was increased in the cryptorchid segment compared with normal. Differently, an opposite trend of expression was shown for ER-beta in the cryptorchid segment compared with normal. Similarly, SOD1 and NRf2 showed a decrease in their expression level in the cryptorchid condition (Figure 4).

In order to highlight any differences in protein expression level between the different segments of the epididymis, we analyzed the data in relation to caput epididymis in both normal and cryptorchid dogs. GPER showed decreased expression along the different segments from the caput to the cauda of the normal dog. ER-alpha and ER-beta showed increased protein level expression in the corpus (Figure 6, panel A). In the cryptorchid epididymis, GPER expression showed an increase in its expression along the epididymal tubule. A similar trend was shown for ER-alpha expression, while ER-beta expression showed a decrease in its expression from the caput to the cauda in the cryptorchid condition (Figure 6, panel A).

### 3.4. mRNA Expression Levels of GPER, ER-Alpha and ER-Beta in the Normal and Cryptorchid Testis–Epididymal Complex: Real-Time RT-PCR

Real-time RT-PCR analysis revealed that all three receptors were expressed in the testis and epididymis of both normal and cryptorchid dogs, albeit with a different trend in expression. In particular, in cryptorchid dogs, mRNA levels increased (for GPER and ER-alpha) in all the segments of the testis–epididymal complex. ER-beta mRNA levels decreased in all segments of the epididymis, while no statistically significant differences were observed in the testis. (Figure 5).

In order to study any differences in mRNA expression level between the different segments of the epididymis, we analyzed the data in relation to caput epididymis in both normal and cryptorchid dogs. GPER and ER-alpha mRNA levels were higher in the cauda than corpus and caput, whereas an opposite trend was shown for ER-beta, with the lowest mRNA levels in the cauda region (Figure 6, panel B). In the cryptorchid epididymis, the mRNA levels of all three receptors were higher in the cauda than the corpus and caput (Figure 6, panel B).

**Figure 6 vetsci-11-00021-f006:**
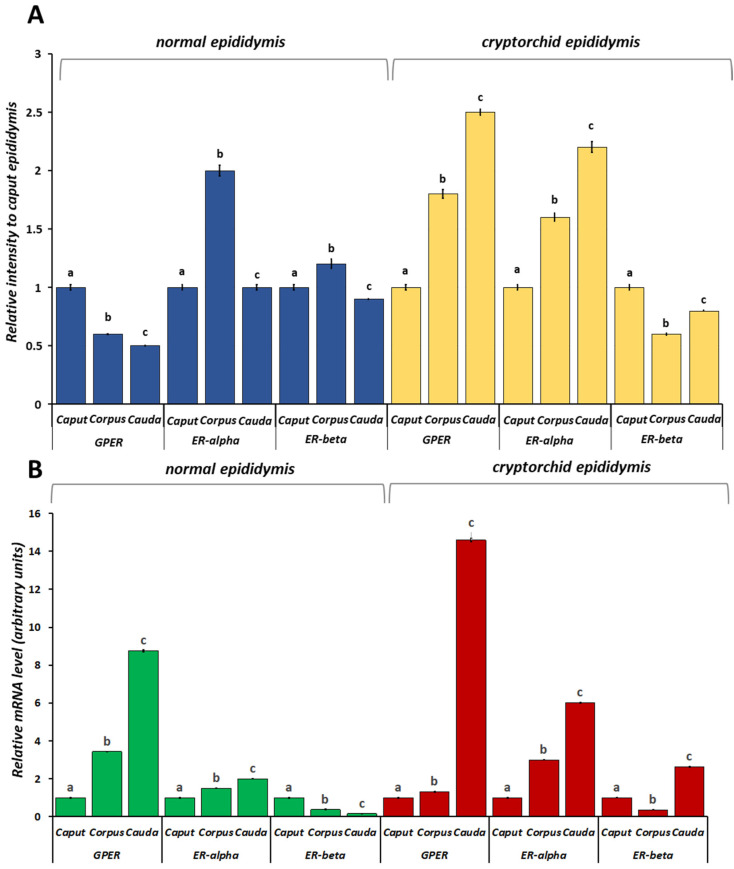
Protein (**A**) and mRNA (**B**) expression levels of GPER, ER-alpha and ER-beta in the epididymal segments of the normal and cryptorchid dog. (**A**) Densitometric analysis: the data are expressed as intensity relative to the caput epididymis. GPER showed decreased expression along the epididymal tubule from the caput to the cauda of the normal dog whereas was increased along the cryptorchid epididymal tract. (**B**) Real-time RT-PCR: the data are expressed as arbitrary units relative to the caput epididymis (the calibrator). In normal dogs, the mRNA expression pattern of GPER and ER-alpha showed a greater increase in the cauda than the corpus and caput, whereas ER-beta showed a greater decrease in the cauda than in the corpus and cauda. In the cryptorchid dog, the mRNA levels of all three receptors were modulated, increasing in the cauda compared with the corpus and caput. The data presented are the mean ± SE of independent experiments. Different letters indicate differences between the examined groups (*p* < 0.05).

## 4. Discussion

In this study, our immunohistochemical results revealed the presence of GPER, ER-alpha and ER-beta in the canine normal testis albeit with a different cellular distribution, suggesting that estrogens probably act through these receptors. In detail, GPER was expressed in both the germ and Leydig cells, and it colocalized with both ER-alpha and ER-beta in the Leydig cells, while it colocalizes only with the last one in the germ cells. Differently, Sertoli cells only expressed ER-beta. In general, the distribution patterns of GPER and ER-alpha were similar, suggesting that estrogens probably act through these receptors. Studies in recent decades have elucidated the roles of estrogen in the physiology of the male reproductive tract, including spermatogenesis regulation and testicular steroidogenesis [14]. GPER mediates predominantly rapid non-genomic signaling of estrogens, and its presence in the testis has been observed in different animal species including horses [23], dogs [31], mice [43], and humans [27,44]. In particular, our results are in accordance with these studies for the localization of GPER and ERs in the Leydig cells, suggesting a role in the modulation of testicular steroidogenesis. The co-existence of multiple receptors in the same cells raises important questions regarding steroid hormone interactions and receptor cross-talk in the control of male reproductive physiology. It was observed that GPER influences Leydig cell morphology and function [43] and can reduce the testosterone secretion in the rat and human Leydig cells [45], suggesting that its dependent non-genomic signaling represents an important mechanism regulating estradiol-dependent steroidogenesis.

The presence of GPER in germ cells (spermatocytes and/or round spermatids) of the normal testis has previously also been confirmed in human testis [27,44], which could suggest its interesting role in spermatogenesis. However, our findings disagree with results from equine [23] and canine testis [31], which showed GPER-IR exclusively in Leydig cells. This discrepancy could be related to different factors, including the diverse sensitivity of the employed antibody and/or fixation methods. The hypothesized role for GPER in germ cells could be participation in the modulation of spermatogonia proliferation and in physiological apoptosis through the regulation of the number of spermatocytes and spermatids. These functions represent two important aspects of spermatogenesis. In particular, the presence of GPER in pachytene spermatocytes and round spermatids could be hypothesized to be related to an activation of the epidermal growth factor receptor/extracellular signal-regulated kinases (EGFR/ERK) pathway, which is involved in the transcriptional modulation of genes controlling apoptosis and differentiation [25,26]. For ERs’ localization, it was reported that ER-alpha and ER-beta are expressed in the testicular germ cells in some animal species including dogs, rodents, and humans, with some differences in their protein distribution [32,34,35,46,47]. According to our findings, GPER is co-expressed only with ER-alpha in round spermatids, suggesting their involvement in the regulation of spermiogenesis, while the localization of ER-beta in Sertoli cells supports germ cell maturation through the supply of different mediators including estrogens, which could be indicative of this function in these cells. Demonstration of this role has been confirmed by the fact that the binding of ERs/estrogens regulates the proliferation and differentiation of immature rat Sertoli cells [24].

In the cryptorchid male gonad, the distribution pattern of GPER also changed with its localization in the Sertoli cells. In detail, early germs cells were negative for all three receptors, while Sertoli and Leydig cells were positive for all. In addition, the protein and mRNA expression level analysis indicated that in the cryptorchid dogs, GPER and ER-alpha increased if compared with normal dogs, while ER-beta decreased. Our results disagree with findings of previous study in horses showing a decreased expression of GPER in cryptorchid animals compared to animals in normal condition [23]. However, interesting evidence obtained by using knockout mice proposes that GPER could be considered a receptor that plays an autonomous role or could cooperate with nuclear estrogen receptors. In addition, the extent to which GPER acts autonomously seems to be related different conditions such as cell type, differentiation status, and pathology, i.e., whether the cell is quiescent, proliferative, or cancerous [48]. The modulation of these receptors suggests that GPER acting with or without the cooperation of canonical ER plays a role in the processes occurring in the cryptorchid condition, including tumorigenesis and/or oxidative stress. In dogs, as in humans, the testis being located in abdominal cavity can result in developmental neoplasia (especially Sertoli cell tumor and seminoma) and torsion [2]. Several studies have demonstrated the involvement of GPER in the tumorigenesis process, as well as its high over-expression in estrogen-dependent tumors in both human [49] and canine testis [31]. In fact, a promoting effect of GPER on cancer progression [22,49] has been proposed, albeit the tumor type could be a factor implicated into this role. This aspect as well as the specific tumor cell type and microenvironment could be relevant for GPER/ER interactions, as demonstrated by the fact that GPER is overexpressed in seminomas but not in non seminomas [50]. As it is known, cryptorchidism can be associated with elevate oxidative status [38,51]. Similarly, our previous results demonstrated a reduced expression of SOD2 in the canine cryptorchid testis, suggesting that high temperature induces oxidative stress and ROS production [16]. This obtained data strengthen our hypothesis, supported also by Kawakami et al. [52] who demonstrated that SOD activity at the testis level was downregulated in (unilateral) cryptorchidism in dogs affected by a Sertoli cell tumor. Elevated intratesticular temperature induces oxidative stress, resulting in apoptosis and impairment of spermatogenesis [53]. ROS are produced physiologically and, in reproduction, participate in capacitation, hyperactivation, acrosome reaction and sperm–oocyte fusion [54]. On the contrary, their excessive production may be detrimental to testis function, they can impair spermatozoa motility and vitality and simultaneously prevent an adequate cellular antioxidant defense system from forming. This condition can lead to an environment unsuitable for normal physiological reactions [55]. In physiological conditions, ROS are maintained at definite levels, and their excesses are eliminated by antioxidant enzymes such as SOD [39]. Recent studies have highlighted the function of Nrf2 as a sensor of oxidative stress in cells, regulating the activity of various antioxidant enzymes to control the redox status at level of cells, and thus preventing oxidative stress [40]. Our Western blot results relative to SOD and Nfr2 are consistent with those demonstrated by Zhao et al. [56] in a study performed on rat testis during aging, wherein decreased activity of SOD and lower Nrf2 expression levels were associated with a reduction in seminiferous tubule diameters and in seminiferous epithelium height. Albeit the investigation was conducted in a different condition, the authors conclude that the observed downregulation of antioxidant ability mediated by Nrf2 pathway could be correlated with the accumulation of oxidative stress; increased DNA damage could be one of the causes implicated in the decline of testicular function during aging. Furthermore, a previous study by Li et al. [57] performed on mouse testis demonstrated high levels of testicular and epididymal lipid peroxidation and associated low levels of antioxidants in the Nrf2 knockout mouse.

In the epithelium of the normal epididymis, the principal cells expressed all three receptors, while basal cells expressed only ER-alpha and ER-beta, except in the cauda, where the basal cells showed only ER-beta-IR. In addition, the peritubular muscular cells of the corpus and cauda were positive only for GPER. These different distribution patterns and the differences in the protein expression among the epididymal segments with a decrease from the caput to the cauda suggest that GPER is involved in the different functions of each epididymal segment, and that different regional abundances of GPER and ERs exist. Our findings present the idea that GPER could take part in the secretory activity of the epithelium with or without the cooperation of ERs. The signaling mechanism activated (triggered) by GPER could be important for the promotion of the secretory activity needed to create an appropriate microenvironment for sperm maturation. As known, estrogens have an important role in maintenance of some aspects of epididymal physiology [14]. The expression of ERs has been widely documented in the epididymis of several species (mice, rats, dogs, cats, and monkeys) [32,35,36,46,58] with a common observation of the presence of ER-alpha in the efferent ductules that not only provide to the reabsorption of more than 90% of the rete testis fluid but concentrate sperm for epididymal storage [59]. In addition, the very wide GPER expression in the epididymis of pigs [28,60], rats [30], sheep [61] and in humans’ ductuli efferents and proximal epididymis [29] suggests its involvement in sperm maturation, protection, and storage. However, GPER could play other different roles such as in the regulation of contractility, demonstrated by its presence in the muscle cells of the human ductuli efferent and proximal epididymis [29,62]. In the cryptorchid epididymis, GPER and ERs have a distribution pattern like the normal epididymis. On the other hand, we observed a change in the expression level of GPER by an increase along epididymal tubule from the caput to cauda. Similar to this pattern was the expression of ER-alpha, while a decrease was observed for ER-beta. In addition, comparing each cryptorchid epididymal segment with normal segments, an increase in the expression level of GPER and ER-alpha was observed, alongside a decrease in ER-beta, suggesting that GPER and ER-alpha could cooperate in their functions. Why does GPER increase in the cryptorchid epididymis? In the cryptorchid condition, spermatogenesis is almost completely blocked, and therefore the functions of the epididymis are also impaired, or in any case deregulated. Therefore, we also observed a change in the morphology of the epithelium and an increase in stromal tissue compared with epithelial tissue. Furthermore, our results showed, in association with these results, a reduction in SOD1 and Nrf2 expression levels, suggesting that an unbalanced oxidative status was present in the cryptorchid epididymis. Our findings are in accordance with a study by Asl et al. [63], in which it was observed that the expression of Nrf2 signaling decreased during cryptorchidism, especially in the epididymis. Based on these considerations, we could hypothesize, although no specific functional assays were performed, that GPER is not directly involved as a receptor in the regulation of epididymal functions, but probably protects against damage from oxidative stress. This hypothesis is derived from literature data showing that estrogens protect from oxidative damage [64]. GPER-activation has been reported to exert protective effects against H_2_O_2_-induced oxidative stress and toxicity in different cell types such as pancreatic cells [65], renal epithelial cells [66], the heart [67,68], and intestinal cryptic cells [69]. As the results of this research are fundamentally based on a protein distribution and expression approach, future specific functional studies will examine specific GPER regulation. Overall, this work is indicative of the complex expression of ERs, specifically in the cryptorchid condition, and the more intricate functions that remain to be elucidated. Moreover, this study constitutes a starting point to guide future work on GPER’s roles other male reproductive diseases.

## 5. Conclusions

Our results describe for the first time the distribution and the expression of GPER in the testis–epididymal complex of cryptorchid dogs. In particular, the increased expression of GPER and ER-alpha along the tracts of cryptorchid epididymis is accompanied by a change in the morphology of the epithelium. These results are the basis for further functional and molecular studies to better characterize the role of GPER in this physiopathological condition.

## Figures and Tables

**Figure 1 vetsci-11-00021-f001:**
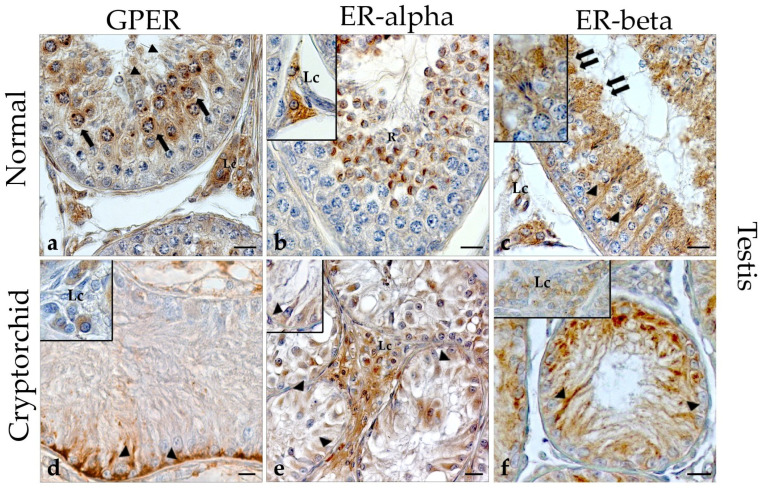
Immunohistochemical staining of GPER, ER-alpha, and ER-beta in the normal and cryptorchid testis of dogs. In the normal testis, germ cells and Leydig cells were positive for all three receptors. In particular, the group of Leydig cells positive for GPER-, ER-alpha and ER-beta are labeled with fine granular stains dispersed through the cytoplasm (Lc); oval spermatids were positive for GPER (line arrow); ER-alpha-IR was found in round/immature spermatids (R); ER-beta-IR was found in a group of Sertoli cells (arrowheads) in which the cytoplasm was accompanied by elongated/mature spermatids (double arrow, insert). In the cryptorchid testis, Leydig (Lc) and Sertoli (arrowheads, insert) cells were positive for all three receptors. Bar: 25 microns.

**Figure 2 vetsci-11-00021-f002:**
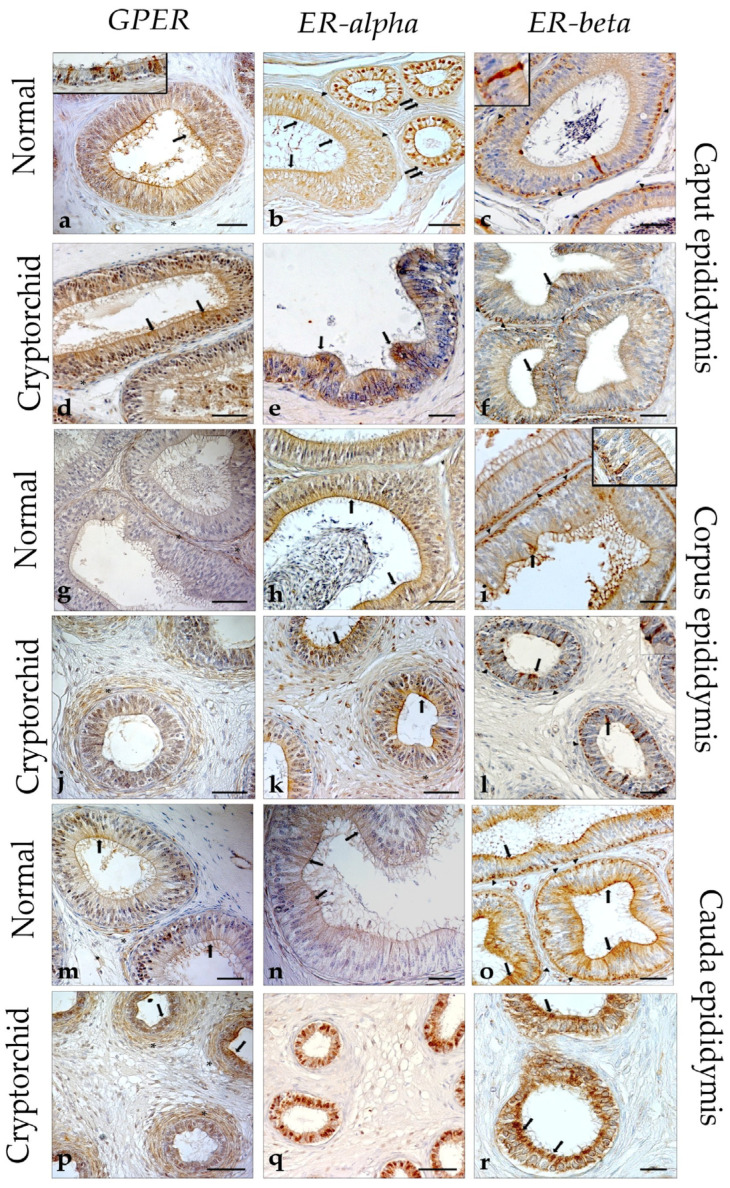
Immunohistochemical staining of GPER, ER-alpha and ER-beta in the normal and cryptorchid epididymis of dogs. In the normal dog, GPER was distributed in the apical portion of the principal epithelial cells, in the relative stereocilia-covered surface of caput (**a**), arrows, detail in insert) and cauda (**m**), arrows, and in the peritubular muscular cells of all epidydimal segments (**a**,**g**,**m**), asterisks. GPER-IR was detected in the ciliated cells of the efferent ductules (**a**), insert. ER-alpha-IR was detected in the apical portion of the principal cells of all segments (**b**,**h**,**n**), arrows, in the basal cells of the caput (**b**), arrowheads, and in the nuclei of ciliated cells and in some nonciliated type B cells containing vacuoles of the efferent ductules (**b**), double arrows. ER-beta positivity was found in the basal cells of all three segments (**c**,**i**,**o**), arrowheads, in the apical portion of the principal cells of corpus and cauda (**i**,**o**), arrows, and in some entire epithelial cells (**c**,**i**), inserts. In the cryptorchid epididymis, GPER-IR was matched to that described for the normal epididymis (**d**,**j**,**p**). ER-alpha-IR was detected in the apical portion of the principal cells of the caput (**e**), arrows and corpus (**k**), arrows, and in the nuclei of epithelial cells of the cauda (**q**). ER-beta-IR was similar to that described for the normal segments (**f**,**l**), with the exception of the cryptorchid cauda, where only the apical portion of the principal cells was positive (**r**), arrows. Bar: 30 microns.

**Figure 3 vetsci-11-00021-f003:**
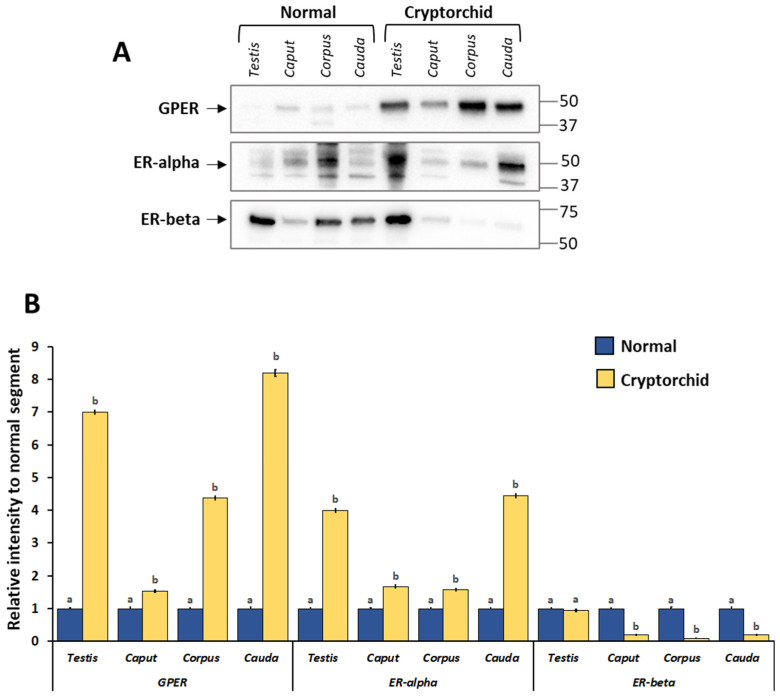
Western blot and densitometric analysis of the GPER, ER-alpha and ER-beta in the testis–epididymal complex of normal and cryptorchid dogs. (**A**) Representative immunoblots. (**B**) The results are expressed as the intensity relative to that for normal tract. GPER and ER-alpha expression increased in the cryptorchid tract compared with the normal, while ER-beta decreased. The data presented are the mean ± standard error (SE) of independent experiments. Different letters indicate differences between the examined groups (*p* < 0.05).

**Figure 4 vetsci-11-00021-f004:**
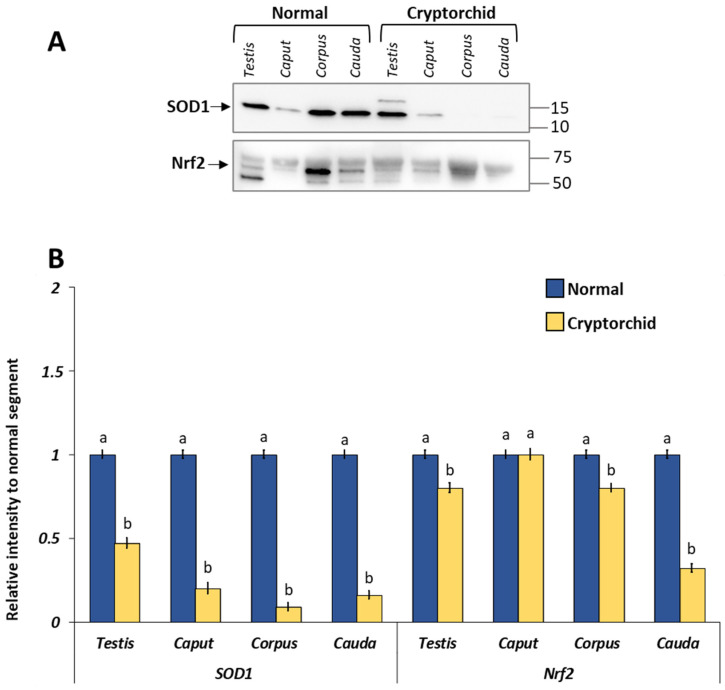
Western blot and densitometric analysis of the SOD1 and Nrf2 in the testis–epididymal complex of normal and cryptorchid dogs. (**A**) Representative immunoblots. (**B**) The results are expressed as intensity relative to that of a normal tract. SOD1 and Nrf2 decreased in the cryptorchid tract compared with the normal. The data presented are the mean ± SE of independent experiments. Different letters show differences between the examined groups (*p* < 0.05).

**Figure 5 vetsci-11-00021-f005:**
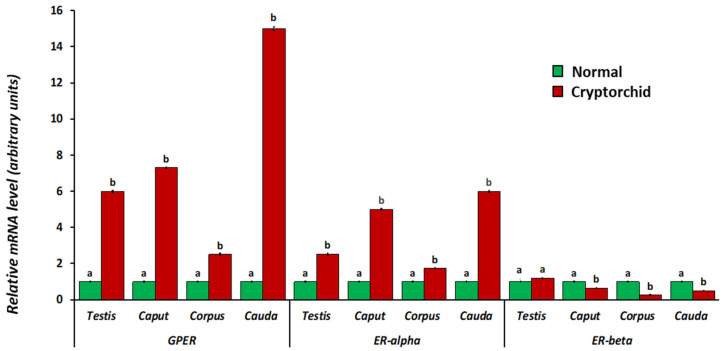
GPER, ER-alpha and ER-beta mRNA level expression found using real-time RT-PCR in the testis–epididymal complex of normal and cryptorchid dogs. The calibrator is the normal segment. The reference gene was GAPDH. The GPER and ER-alpha mRNA levels increased in the cryptorchid segment. Conversely, ER-beta mRNA levels decreased in the cryptorchid segments. The data presented are the mean ± SE of independent experiments. Differences between the examined groups (*p* < 0.05) are indicated by different letters.

**Table 1 vetsci-11-00021-t001:** List of primers pairs that were used for mRNA analysis in real-time RT-PCR.

Primer Name	Gene Name	Primer Sequence 5′-3′	Genbank Accession Number	Product Size
dGPERfor dGPERrev	*Canis lupus familiaris* GPER1	AAAGCCTGCAGTGTCTTGGTATC TGGGTACTGGTGATTCTGGACTT	XM_005621204.2	150 bp ^1^
dESR1for dESR1rev	*Canis lupus familiaris* ESRA	TCGGAAAACTGCTCCTGTAAATG ACCACAATCTCTCGGTCAAAGAG	NM_001002936.1	150 bp
dESR2for dESR2rev	*Canis lupus familiaris* ESRB	CGTGCTAGAGATGAAATCGTTAATG CCCCTGTTTCCTGAGCAGTCTAT	XM_038591983.1	152 bp
dGAPDHfor dGAPDHfor	*Canis lupus familiaris* GAPDH	TGTCCCCACCCCCAATG TCGTCATATTTGGCAGCTTTCTC	XM_003434387	69 bp

^1^ bp: base pairs.

**Table 2 vetsci-11-00021-t002:** Immunostaining pattern of GPER, ER-alpha and ER-beta in the testis of normal and cryptorchid dogs.

				Cytotypes			
	Leydig Cells	Sertoli Cells	Pre-Meiotic Cells	Pachytene Spermatocytes	Round Spermatids	Oval Spermatids	Elongated Spermatids
**GPER**							
Normal	+++	-	-	+++	+	++	-
Cryptorchid	++	++	-	-	-	-	-
**ER-alpha**							
Normal	++	-	-	-	+++	-	-
Cryptorchid	+++	++	-	-	-	-	-
**ER-beta**							
Normal	++	+++	-	-	-	-	+++
Cryptorchid	+++	+++	-	-	-	-	-

+++ = high intensity; ++ = moderate intensity; + = low/mild intensity; - = absence (intensity of staining).

**Table 3 vetsci-11-00021-t003:** Immunostaining pattern of GPER, ER-*alpha* and ER-*beta* in the epididymis of normal and cryptorchid dogs.

			GPER		ER-Alpha			ER-Beta		
	Epididymal Segments			Immunostaining Pattern						
		P	B	PM	P	B	PM	P	B	PM
NORMAL	Caput	+	-	-	++	+++	-	++	+++	-
	Corpus	++	-	+	++	-	-	++	+++	-
	Cauda	++	-	+	++	-	-	+++	+++	-
	Caput	++	-	-	++	-	-	++	+++	-
CRYPTORCHID	Corpus	+	-	++	++	-	-	+	++	-
	Cauda	++	-	+++	+ *	-	-	+++	-	-

P = principal cells; B = basal cells; PM = peritubular muscular cells; +++ = high intensity; ++ = moderate intensity; + = low/mild intensity; +/- = rare; * = nuclear signal; - = absence (intensity of staining).

## Data Availability

Data are contained within the article and Appendix A.

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
