# Peer review of "G Protein-Coupled Estrogen Receptor (GPER) and ERs Are Modulated in the Testis–Epididymal Complex in the Normal and Cryptorchid Dog"

_vetsci, 2024, doi:10.3390/vetsci11010021_

Round 1

Reviewer 1 Report

Comments and Suggestions for Authors

Veterinary Sciences

Manuscript Number: vetsci-2719300

“GPER is modulated in the testis-epididymal complex in the normal and cryptorchid dog” by G. Liguori et al.

This manuscript, combining Western blot, qRT-PCR, and immunohistochemistry investigated the abundance and distribution pattern of GPER. ER-alpha and ER-beta in the testis and epididymis of  normal and cryptorchid dogs.

The research is interesting and contributes to our knowledge of the molecular factors underlying the male genital apparatus in normal and pathological conditions.

Particularly, the results indicate that the cryptorchid condition influences the expression of GPER, ER-alpha, and ER-beta in the testis and epididymis of dogs, thus evidencing possible implications of these receptors in the testis and epididymis functions.

 The MS cannot be published in the following form but needs major revision.

Below I give suggestions for improving the quality of the paper.

Materials and Methods

Line 121: list the steps of the tissue preparation;

Line 126: Why was the method of reference 40 followed instead of methods for immunolocalization of GPER, ER-alpha, and ER-beta?

Line 142: specify what “image documentation” means.

Results

Lines 218-224: the immunolocalization results of ER-alpha and ER-beta in the testis must be described separately.    

Lines 223-224: the granule present in the tail of the elongated spermatids is not clearly distinguishable in Figure 5. Please, show it in a higher magnification inset.

Line 226: in Fig. 1d the arrowheads of the left seminiferous tubule show peritubular cells but no Sertoli cells. A higher magnification picture would show what the authors claim.

Lines 226-227: specify what “early germ cells” means.

Line 229: The authors state that the cytoplasm of the Sertoli cells in Fig. 1f contains intensely stained granules. Unfortunately, they are not distinguishable in this picture.  A picture at higher magnification would show what the authors state.

Line 238: Figure 1. Unlike what is reported in the results section and Table 2, no positivity of Sertoli cells is distinguishable in pictures 1d and 1e.

Line 240: replace "semi-quantitative localization" with "localization and semi-quantitative evaluation".

Line 242: Table 2. Replace “localization” with “Immunostaining pattern”.

Line 243: Table 2. Specify what "density" means. Do the authors refer to the number of cells per unit area or the staining intensity?

Line 246: The dog efferent ductules present two types of cells: some having very prominent cilia and others possessing only microvilli (Chandler et al., 1981).  Specify the GPER-positive cell type in the efferent ductules.

Line 248-249: the authors claim to observe GPER positivity in the brush border of the epithelial cells of the caput and cauda segments.  The brush border is the microvillus-covered surface. Ultrastructural investigations (Schimming and Vicentini, 2001s ) displayed that the apical surface of the dog epididymis has stereocilia but not microvilli. Therefore it is not correct to speak of a "brush border".

Line 250: “Figure 2g,m – asterisk”. The asterisk is also present in picture "a". Was the asterisk put in "a"  by mistake or should it be added in line 250?

Line 252: the authors stated, “ER-alpha-IR was found in the epithelial cells of the efferent ductule (nuclear signal)”. Which cell type is ER-alpha positive? The positivity is also visible in the cytoplasm of some epithelial cells. Please, add this in the text.

Line 254: Figure 2n. Contrary to what the authors state, no clear signals of apical positivity are visible in picture "n".

Lines 255-257: the sentence " some positive epithelial cells completely filled by condensed granules which defined the entire profile of the cells were found in all epididymal segments" should be rewritten as follows "some positive cells were enterely positive from their basal zone to the apical zone”. However, ER-beta positivity is visible in the apical region of all epididymal segments and especially in the cauda”.

Line 261: Because of the height of the epithelium, the inset in Figure a is a detail of the efferent ducts. Edit caption.

Line 275: are you referring to Table 1 or Table 3?

Line 276: replace 2k and 2q with 2j and 2p, respectively;

Line 277: delete "the number of positive structures was higher than normal segment (Figure 2)." because the cell count was not carried out in this study.

Table 3: why was staining intensity not reported in Table 3 as well as in Table 2?

Table 3: as for the ductuli efferent immunostaining, specify whether the immunoreactivity was found in ciliated or nonciliated cells (Chandler et al., 1981). 

Lines 327-328: correct the statement "in all the segments of the testis-epididymal complex (Figure 5)" because Figure 5 shows no significant difference between cryptorchid and normal testes.

Discussion

Lines 391-392: the sentence " In our findings GPER is co-expressed only with ER-alpha in germ cells suggesting their involvement in the regulation of spermatogenesis" is very general and incorrect because GPER and ER-alpha are both expressed only in round spermatids and not in all germ cells (see Table2). Therefore, this part needs to be rewritten.

Line 425: add the reference number after “Kawakami et al”.

Line 439: the statement that the basal cells expressed both ER-alpha and ER-beta is not accurate because the basal cells of the caudal region do not show ER-alpha (see Table 3).  Therefore this part needs to be rewritten.

Lines 458-459: the statement " In the cryptorchid epididymis, GPER and ERs have a distribution pattern like that of normal epididymis" is not quite correct because the basal cells of the cauda epididymis do not express ER-beta (Table 3). This issue should be discussed.

Line 471: add the reference number after “Asl et al.”

Author Response

Reviewer# 1:

This manuscript, combining Western blot, qRT-PCR, and immunohistochemistry investigated the abundance and distribution pattern of GPER. ER-alpha and ER-beta in the testis and epididymis of normal and cryptorchid dogs.

The research is interesting and contributes to our knowledge of the molecular factors underlying the male genital apparatus in normal and pathological conditions.

Particularly, the results indicate that the cryptorchid condition influences the expression of GPER, ER-alpha, and ER-beta in the testis and epididymis of dogs, thus evidencing possible implications of these receptors in the testis and epididymis functions.

The MS cannot be published in the following form but needs major revision

Author's Notes to Reviewer 1

We wish to thank you for the time spent on our manuscript (Manuscript Number: vetsci-2719300) entitled “GPER is modulated in the testis-epididymal complex in the normal and cryptorchid dog” and for the opportunity to submit a revised version of the manuscript modified in response to your comments.

Below you will find a point-by-point answer to all remarks and changes we made in the manuscript. The changes in the sentences of the manuscript were made by ticking the original text and by adding the new text. Every change was highlighted in red.

In Bold are reported the Reviewers comments.

Below I give suggestions for improving the quality of the paper.

Materials and Methods

Line 121: list the steps of the tissue preparation;

Author response: As suggested by the reviewer, we modified the sentence to describe the step of tissue preparation (see lines 137-139).

Line 126: Why was the method of reference 40 followed instead of methods for immunolocalization of GPER, ER-alpha, and ER-beta?

Author response: As kindly suggested by the reviewer, we provided to modify the sentence and eliminated the reference 40 (see lines 144-145).

Line 142: specify what “image documentation” means.

Author response: We thank the reviewer for the kind suggestion, we provided to replace “image documentation” with “image capture” in order to better specify what “image documentation” means (see line 159).

Lines 218-224: the immunolocalization results of ER-alpha and ER-beta in the testis must be described separately.   

Author response: As suggested by the reviewer, we added separately the immunolocalization results of ER-alpha and ER-beta in the testis (see lines 244-253).

Lines 223-224: the granule present in the tail of the elongated spermatids is not clearly distinguishable in Figure 5. Please, show it in a higher magnification inset.

Author response: We thank the reviewer for the kind suggestion, we added a higher magnification insert.

Line 226: in Fig. 1d the arrowheads of the left seminiferous tubule show peritubular cells but no Sertoli cells. A higher magnification picture would show what the authors claim.

Author response: As kindly suggested by the reviewer, we provided to replace Fig. 1d with another one and by adding an insert of Leydig cells.

Lines 226-227: specify what “early germ cells” means.

Author response: We thank the reviewer for his/her precise comment and as suggested, we added in the manuscript gonocytes in order to specify what “early germ cells” means.

Line 229: The authors state that the cytoplasm of the Sertoli cells in Fig. 1f contains intensely stained granules. Unfortunately, they are not distinguishable in this picture. A picture at higher magnification would show what the authors state.

Author response: As kindly suggested by the reviewer, we replaced Fig. 1f with another one at higher magnification and by adding an insert of Leydig cells.

Line 238: Figure 1. Unlike what is reported in the results section and Table 2, no positivity of Sertoli cells is distinguishable in pictures 1d and 1e.

Author response: As kindly suggested by the reviewer, we provided to substitute Figs. 1d with another one at higher magnification and to add an insert in Fig 1.d in order to better identify Leydig cells and to add another insert of Sertoli cells in Fig. 1.e

Line 240: replace "semi-quantitative localization" with "localization and semi-quantitative evaluation".

Author response: We thank the reviewer for the comment, and we replaced "semi-quantitative localization" with "localization and semi-quantitative evaluation" (see line 269).

Line 242: Table 2. Replace “localization” with “Immunostaining pattern”.

Author response: We appreciate the punctualization of the reviewer and his/her comment, and we replaced “localization” with “Immunostaining pattern” (Table 2) (see line 271).

Line 243: Table 2. Specify what "density" means. Do the authors refer to the number of cells per unit area or the staining intensity?

Author response: We appreciate the punctualization of the reviewer and according to this suggestion, we provided to specify the “density” means in the manuscript referring to the number of cells per unit area (see line 278).

Line 246: The dog efferent ductules present two types of cells: some having very prominent cilia and others possessing only microvilli (Chandler et al., 1981).  Specify the GPER-positive cell type in the efferent ductules.

Author response: We thank the reviewer for the kind suggestion, and, as reported by Chandler et al.,1981, we specified the GPER-positive cell type by identifying with ciliated cells in text and in the relative caption (see lines 282, 297).

Line 248-249: the authors claim to observe GPER positivity in the brush border of the epithelial cells of the caput and cauda segments. The brush border is the microvillus-covered surface. Ultrastructural investigations (Schimming and Vicentini, 2001s) displayed that the apical surface of the dog epididymis has stereocilia but not microvilli. Therefore, it is not correct to speak of a "brush border".

Author response: As suggested by the reviewer, we provided to replace the “brush border” with “microvillus-covered surface” in the manuscript (see line 276).

Line 250: “Figure 2g,m – asterisk”. The asterisk is also present in picture "a". Was the asterisk put in "a" by mistake or should it be added in line 250?

Author response: As kindly suggested, we provided to replace “Figure 2g,m – asterisk” with “Figure 2a,g,m – asterisk” (see line 284).

Line 252: the authors stated, “ER-alpha-IR was found in the epithelial cells of the efferent ductule (nuclear signal)”. Which cell type is ER-alpha positive? The positivity is also visible in the cytoplasm of some epithelial cells. Please, add this in the text.

Author response: We thank the reviewer for the kind suggestion, and we specified and added the following sentence: “ER-alpha-IR was found non ciliated cells (nuclear signal) and in some type B cells of the efferent ductules”  in the text and in the caption (see line 288, 299).

Line 254: Figure 2n. Contrary to what the authors state, no clear signals of apical positivity are visible in picture "n".

Author response: We thank the reviewer for the kind suggestion, we replaced the Figure 2n with another one with more clear signals of apical positivity.

Lines 255-257: the sentence " some positive epithelial cells completely filled by condensed granules which defined the entire profile of the cells were found in all epididymal segments" should be rewritten as follows "some positive cells were enterely positive from their basal zone to the apical zone”. However, ER-beta positivity is visible in the apical region of all epididymal segments and especially in the cauda”.

Author response: We thank the reviewer for the comment and modified the sentence with "some positive cells were entirely positive from their basal zone to the apical zone” (see line 291-292).

Line 261: Because of the height of the epithelium, the inset in Figure a is a detail of the efferent ducts. Edit caption.

Author response: As kindly suggested by the reviewer, we edited the caption.

Line 275: are you referring to Table 1 or Table 3?

Author response: We thank the reviewer for the accuracy in the revision process. Probably there was carelessness in checking the manuscript. We refer to Table 3.

Line 276: replace 2k and 2q with 2j and 2p, respectively;

Author response: We thank the reviewer for the accuracy in the revision process. We replaced 2k and 2q with 2j and 2p, respectively.

Line 277: delete "the number of positive structures was higher than normal segment (Figure 2)." because the cell count was not carried out in this study.

Author response: As kindly suggested by the reviewer, we provided to delete the sentence "the number of positive structures was higher than normal segment (Figure 2)."

Table 3: why was staining intensity not reported in Table 3 as well as in Table 2?

Author response: As kindly suggested by the reviewer, we modified Table 3 by introducing the density parameter.

Table 3: as for the ductuli efferent immunostaining, specify whether the immunoreactivity was found in ciliated or non ciliated cells (Chandler et al., 1981).

Author response: We thank the reviewer for the kind suggestion, we deleted the part relative to efferent ductules cytotypes in order to avoid misunderstandings.

Lines 327-328: correct the statement "in all the segments of the testis-epididymal complex (Figure 5)" because Figure 5 shows no significant difference between cryptorchid and normal testes.

Author response: We thank the reviewer for the kind suggestion, we modified the statement in the manuscript with this sentence “ER-beta mRNA levels decreased in all segments of the epididymis while no statistically significant differences were observed in the testis” (see lines 376-378)

Discussion

Lines 391-392: the sentence "In our findings GPER is co-expressed only with ER-alpha in germ cells suggesting their involvement in the regulation of spermatogenesis" is very general and incorrect because GPER and ER-alpha are both expressed only in round spermatids and not in all germ cells (see Table2). Therefore, this part needs to be rewritten.

Author response: According to the suggestion by the reviewer that we thank for his/her careful comment, the sentence was changed in: “In our findings GPER is co-expressed only with ER-alpha in round spermatids suggesting their involvement in the regulation of spermiogenesis” (see lines 453-454).

Line 425: add the reference number after “Kawakami et al”.

Author response: We thank the reviewer for this suggestion, and we moved the reference number (now renamed as 52) after “Kawakami et al” (see line 484).

Line 439: the statement that the basal cells expressed both ER-alpha and ER-beta is not accurate because the basal cells of the caudal region do not show ER-alpha (see Table 3).  Therefore, this part needs to be rewritten.

Author response: As suggested by the reviewer, we modified the part by adding the following sentence: “except the cauda where the basal cells showed only ER-beta-IR” (see line 506).

Lines 458-459: the statement "In the cryptorchid epididymis, GPER and ERs have a distribution pattern like that of normal epididymis" is not quite correct because the basal cells of the cauda epididymis do not express ER-beta (Table 3). This issue should be discussed.

Author response: We thank the reviewer for the suggestion, we modified the sentence as follows:” In the cryptorchid epididymis, GPER and ERs have a distribution pattern similar to the normal epididymis”. We have focused our attention on the change in the morphology of the epithelium and an increase in stromal tissue compared to the epithelial one. This aspect led us to hypothesize that GPER is not directly involved as a receptor in the regulation of epididymal functions, but probably plays a protective role against damage of oxidative stress (see line 524-525).

Line 471: add the reference number after “Asl et al.”

Author response: We thank the reviewer for this suggestion, and we moved the reference number (now renamed as 63) after “Asl et al.” (see line 537).

Reviewer 2 Report

Comments and Suggestions for Authors

Dear Authors,

thanks for this interesting paper,

I have only a  about Lines  50-51 and 54, the meaning is not clear

Author Response

Author's Notes to Reviewer# 2

We wish to thank you for the time spent on our manuscript (Manuscript Number: vetsci-2719300) entitled “GPER is modulated in the testis-epididymal complex in the normal and cryptorchid dog” and for the opportunity to submit a revised version of the manuscript modified in response to your comments.

Below you will find the answer to your remarks and changes we made in the manuscript. The changes in the sentences of the manuscript were made by ticking the original text and by adding the new text. Every change was highlighted in red.

In Bold are reported the Reviewers comments.

I have only a about Lines 50-51 and 54, the meaning is not clear

Author response: We thank the reviewer for this suggestion, and we changed the sentence reported in lines 50-51 with “this condition is common also in other species (stallion, boar and human)” (see line 55). For the line 54, the sentence has been changed in: “The cryptorchidism is associated to morphological by structural and functional anomalies of tubular and interstitial compartments of testis and epididymis [7,8] due to the fact that the undescended testis exposed to body temperature, higher that scrotal one, may impair the sperm production (see lines 55-61). 

Reviewer 3 Report

Comments and Suggestions for Authors

The present article aims to compare the expression of estrogen receptors in the testicular and epididymal epithelium of cryptorchid and normal dogs, as well as the expression of antioxidant enzymes in the testicle and epididymis. For this purpose, testis and epididymis of cryptorchid and adult male dogs were subjected to immunohistochemical analysis and mRNA expression of estrogen receptors (ERs) alpha and beta, and a transmembrane ER, namely GPER (G-protein-coupled ER 1), in addition to the evaluation of the protein expression of estrogen receptors (ER-α, ER-β and GPER) and antioxidant enzymes (SOD and Nrf2).

This is a very well-written scientific article that provides a robust foundation based on previous experiments and scientific literature, in addition to presenting novel data on the gene expression of estrogen receptors in different epididymal segments. This is an important area of investigation, and indeed, research on a deeper understanding of cryptorchidism in dogs is still limited. Overall, some interesting comparisons may prove fruitful in future studies. However, there are some points of concern that need to be clarified before a comprehensive evaluation of the article for publication.

Some of the major comments and criticisms include:

Title: The title of the work does not accurately reflect what was actually carried out in the experiment. It would be beneficial to avoid using the abbreviation for the GPER receptor and to include the other receptors and the redox study.

Simple summary: All these information has to be useful for non-researchers. Therefore, the terminology is not adequate.

Line 24: and what about the expression of ER alpha and beta?

Line 26: these genes were not previously referred in the methodology and should not stand alone at this point in the summary.

Summary:

Lines 41: have you indeed evaluated the expression of SOD and Nrf2? Please, clarify

Lines 41-42: this is speculative for a conclusion. Please, rewrite.

Introduction:

More information should be provided regarding the mechanisms involved on the etiology, pathogenesis and consequences of cryptorchidism in dogs.

Line 52-53: Please clarify whether this relationship is a cause or consequence of testicular ectopia.

Lines 88-89: it is not clear by now, why do you expect to see a relationship between the expression of estrogen receptors and antioxidant enzymes. This needs be better clarified.

Lines 96-97: please make a more in-depth description of the relationship between SOD and Nrf2

Material and methods:

Please, describe in detail the subjects you used to compose the groups. What was, in fact, the experimental subjects (dogs or testicles)? Did the same dogs simultaneously belong to both the control group and the cryptorchidism group? In other words, do the animals exhibit both the disease and normality, i.e., the scrotal testicles were used for the control group and the ectopic testicles for the cryptorchid group?

Were there any macroscopic sign of testicular degeneration or further atrophy? In the same manner, did testicles present any sign of neoplasia? Please include a macroscopic description of both ectopic and scrotal testicles. 

Line 120: this is confusing. The groups had 10 dogs each, that is correct?

Lines 125-144: Did you perform the immunohistochemical analysis of SOD and Nrf2? Did you perform any positive control of the immunohistochemistry? Please describe the methodological data analysis of immunohistochemistry? quantitatively or semi-quantitatively?

Line 199: Standard deviation stands for the measure of dispersion of the data from the mean. Conversely, standard error (SE) is a precision of the means or in comparing and testing differences between means. Thus, it is recommendable to present your data as: means ± SE.

Results:

Lines 217-218: This result is surprising, considering that the Sertoli cell is the primary steroidogenic cell of the seminiferous epithelium.

Discussion

Lines 398-399: Did the seminiferous tubules of ectopic testicles show a higher quantity of immature sperm lineage compared to the testicles in the control group? Moreover, is it possible that the ongoing testicular degeneration process hindered the progression of spermatogenesis? Please clarify.

My main concern regarding the discussion of data from ectopic testicles is the exclusion of an ongoing testicular degeneration or neoplastic formation, both common in cryptorchidism. Thus, it cannot be conclusively stated that the differentiated expression of estrogen receptors is a consequence of cryptorchidism, but rather a result of testicular injury due to the ectopia of the testicles.

Lines 420-421: this is overstated because you have not performed a histological analysis of the ectopic testicles in order to affirm an ongoing neoplastic transformation.

Lines 433-437: Why have you neglected your results of western blot SOD and Nfr2? Try to make a more tightened connection between your results and the literature ones. 

Conclusion

The conclusion is not aligned with the study's objectives. Try to adhere more directly to your data and refrain from making statements that were not directly observed in the present study. The conclusion needs to be rewritten.

Author Response

Reviewer# 3

EVALUATION

The present article aims to compare the expression of estrogen receptors in the testicular and epididymal epithelium of cryptorchid and normal dogs, as well as the expression of antioxidant enzymes in the testicle and epididymis. For this purpose, testis and epididymis of cryptorchid and adult male dogs were subjected to immunohistochemical analysis and mRNA expression of estrogen receptors (ERs) alpha and beta, and a transmembrane ER, namely GPER (G-protein-coupled ER 1), in addition to the evaluation of the protein expression of estrogen receptors (ER-α, ER-β and GPER) and antioxidant enzymes (SOD and Nrf2).

This is a very well-written scientific article that provides a robust foundation based on previous experiments and scientific literature, in addition to presenting novel data on the gene expression of estrogen receptors in different epididymal segments. This is an important area of investigation, and indeed, research on a deeper understanding of cryptorchidism in dogs is still limited. Overall, some interesting comparisons may prove fruitful in future studies. However, there are some points of concern that need to be clarified before a comprehensive evaluation of the article for publication.

Author's Notes to Reviewer#3

We wish to thank you for the time spent on our manuscript (Manuscript Number: vetsci-2719300) entitled “GPER is modulated in the testis-epididymal complex in the normal and cryptorchid dog” and for his/her important comments and suggestions that certainly help us to improve the quality of the manuscript.

My co-authors and I have gone through your comments carefully and tried our best to address them one by one. We hope the manuscript has been improved accordingly.

Below you will find a point-by-point answer to all remarks and changes we made in the manuscript. The changes in the sentences of the manuscript were made by ticking the original text and by adding the new text. Every change was highlighted in red.

In Bold are reported the Reviewers comments.

Some of the major comments and criticisms include:

Title: The title of the work does not accurately reflect what was actually carried out in the experiment. It would be beneficial to avoid using the abbreviation for the GPER receptor and to include the other receptors and the redox study.

Author response: We thank the reviewer for his/her suggestion. We added the entire name for the GPER receptor and also the ERs. Regarding the suggestion to include in the title also the redox study, we think; 1) that the modulation of GPER expression and the study on ERs is the more relevant and innovative result from our research study, 2) we would maintain a concise title also considering the specific instructions of the Journal for the title (concise, specific and relevant).

Simple summary: All these information has to be useful for non-researchers. Therefore, the terminology is not adequate.

Line 24: and what about the expression of ER alpha and beta?

Author response: We appreciate the reviewer for his/her suggestion, and we modified the simple summary by introducing results referring to the expression of ER alpha and beta (lines 29-31).

Line 26: these genes were not previously referred in the methodology and should not stand alone at this point in the summary.

Author response: We thank the reviewer for his/her suggestion. In order to complete the description of the aim of the research, we added a sentence describing more detailed the significance of SOD1 and Nrf2 for the research purposes. “In addition, in these tissues the expression level of two proteins as SOD1 and Nrf2 normally associated to oxidative stress was investigated to evaluate possible relation with ERs (lines 26-28).

Summary:

Lines 41: have you indeed evaluated the expression of SOD and Nrf2? Please, clarify.

Author response: As precisely suggested by the reviewer, we added a sentence describing the evaluation of SOD and Nrf2 (see lines 41-42) prior to discuss about the relative results obtained.

Lines 41-42: this is speculative for a conclusion. Please, rewrite.

Author response: According to the suggestion by the reviewer, we modified the conclusion (see lines 46-48).

Studies involving animal subjects’ paragraph:

The presented statement is incorrect. As long as the studies are conducted on animals or animals’ tissues, the experiments involve animal subjects.

Author response: We appreciate the comment by the reviewer, and we modified the title of the sub-section in “Animals” (line 115) and added another sub-section entitled “2.2 Tissue collection” (line 129).

Introduction:

More information should be provided regarding the mechanisms involved on the etiology, pathogenesis and consequences of cryptorchidism in dogs.

Author response: We appreciate the accurate suggestion by the reviewer, and we added more detailed information to better describe mechanisms involved on the etiology, pathogenesis and consequences of cryptorchidism in dogs (see lines 55-57).

Line 52-53: Please clarify whether this relationship is a cause or consequence of testicular ectopia.

Author response: We thank the reviewer for his/her comment, and we added comments to clarify this concept. The sentence was changed with “The cryptorchidism is accompanied associated to morphological by structural and functional alterations anomalies of tubular and interstitial compartments of testis and epididymis [7,8] due to the fact that the undescended testis exposed to body temperature higher that scrotal one may impair the sperm production as principal consequences of impaired spermatogenesis and germ cell damage [9, 10].” (see lines 58-61).

Lines 88-89: it is not clear by now, why do you expect to see a relationship between the expression of estrogen receptors and antioxidant enzymes. This needs be better clarified.

Author response: We thank the reviewer for his/her comment. As suggested, we introduced sentences to describe the possible relationship between the expression of estrogen receptors and antioxidant enzymes.  “Estrogens through its receptors may be involved in the regulation of biological processes (cellular proliferation, metabolic activity and reproduction) For this reason, estrogen-stress interactions were considered to understand the antioxidant action on tissues that perform reproductive functions (ref. Niranjan M.K. et al.,  Expression of estrogen receptor alpha in response to stress and estrogen antagonist tamoxifen in the shell gland of Gallus gallus domesticus: involvement of anti-oxidant system and estrogen. Stress 2021, 24, 3, 261–272 https://doi.org/10.1080/10253890.2019.1710127)  (see lines 97-100).

Lines 96-97: please make a more in-depth description of the relationship between SOD and Nrf2

Author response: We thank the reviewer for his/her accurate comment. Several studies demonstrate a relationship between the SOD1 enzyme and the Nrf2 antioxidant system. In particular, Kirby et al. demonstrated that the presence of mutSOD1 in a mouse motor neuron-like cell line resulted in reduced Nrf2 mRNA expression and downregulation of Nrf2 target genes (Kirby, E. Halligan, M. J. Baptista et al., Mutant SOD1 alters the motor neuronal transcriptome: implications for familial ALS", Brain, vol. 128, no. 7, pp. 1686–1706, 2005) (see lines 108-111).

Materials and Methods:

Please, describe in detail the subjects you used to compose the groups. What was, in fact, the experimental subjects (dogs or testicles)? Did the same dogs simultaneously belong to both the control group and the cryptorchidism group? In other words, do the animals exhibit both the disease and normality, i.e., the scrotal testicles were used for the control group and the ectopic testicles for the cryptorchid group?

Author response: Regarding the observation by the reviewer, we modified the text in “The current research was performed on the complex testis-epididymis obtained from a total of n.20 adult male mixed breed dogs divided into two groups: control (n = 10) and cryptorchid (n = 10). In detail, the control group was identified by mature healthy dogs (average weight 19.8 ± 2.7 kg, average age = 4.8 ± 1.91 years), while the cryptorchid group was represented by dogs with unilateral cryptorchidism (testis retained in the abdomen/in inguinal canal, average weight). (lines 116-120).

Each group (both the normal and the cryptorchid) was then divided into two subgroups of 5 dogs each. One subgroup was used for immunohistochemistry studies and the other subgroup was used for molecular biology (Western blot and real-time-RT-PCR). 

Were there any macroscopic sign of testicular degeneration or further atrophy? In the same manner, did testicles present any sign of neoplasia? Please include a macroscopic description of both ectopic and scrotal testicles.

Author response: We appreciate the accurate suggestion by the reviewer, and we added more detailed information to better describe the macroscopic morphology of both ectopic and scrotal testis (see lines 131-133) in material and methods section. In addition, cryptorchid dogs that had symptoms attributable to hormonal imbalances associated with testicular tumors were excluded from the study and cryptorchid testes that showed macroscopic lesions attributable to neoplasms upon sampling were excluded.

Line 120: this is confusing. The groups had 10 dogs each, that is correct?

Author response: Regarding this comment by the reviewer, as we reported in a previous comment, we added a sentence; “Each group (both the normal and the cryptorchid) was then divided into two subgroups of 5 dogs each. One subgroup was used for immunohistochemistry studies and the other subgroup was used for molecular biology (Western blot and real-time-RT-PCR)” (see lines 136-141). 

Lines 125-144: Did you perform the immunohistochemical analysis of SOD and Nrf2? Did you perform any positive control of the immunohistochemistry? Please describe the methodological data analysis of immunohistochemistry? quantitatively or semi-quantitatively?

Author response: We thank the reviewer for his/her comment. Regarding the first question, we did not perform the immunohistochemical analysis of SOD and Nrf2. The focus of the research is GPER and ERs. Relatively to the second question the analysis of immunohistochemistry was semi-quantitative and were performed by two different blind observers.

In detail, each observer made an image evaluation considering the density parameter as the number of cells per unit area. On the basis of this semi-quantitative evaluation, the following scores were awarded:

+++ = high density; ++ = moderate density; + = low/mild density; +/=rare;  - = absence

Line 199: Standard deviation stands for the measure of dispersion of the data from the mean. Conversely, standard error (SE) is a precision of the means or in comparing and testing differences between means. Thus, it is recommendable to present your data as: means ± SE.

Author response: As suggested by the reviewer that we thank for his/her careful suggestion, we modified the presentation of the data as means ± SE and therefore, we modified the Figures (see line 223).

Results:

Lines 217-218: This result is surprising, considering that the Sertoli cell is the primary steroidogenic cell of the seminiferous epithelium.

Author response: Regarding this point indicated by the reviewer, we could underlie that similar interesting results were observed by a Walczak-Jedrzejowska R. et al., 2022 (Biology 2022, 11, 373. https://doi.org/10.3390/biology11030373T), albeit in a different pathological condition affecting testis function. The authors show difference in GPER distribution between Sertoli Cells and Leydig cells (between two conditions (complete and aberrant spermatogenesis) showing that Sertoli cells respect to Leydig cells show an increase of GPER distribution and leaving to conclude suggesting two hypotheses to further investigate for GPER at this level (a marker of Sertoli cells maturational state, or involvement in the pathogenesis of observed disturbances).  

Discussion

Lines 398-399: Did the seminiferous tubules of ectopic testicles show a higher quantity of immature sperm lineage compared to the testicles in the control group? Moreover, is it possible that the ongoing testicular degeneration process hindered the progression of spermatogenesis? Please clarify.

My main concern regarding the discussion of data from ectopic testicles is the exclusion of an ongoing testicular degeneration or neoplastic formation, both common in cryptorchidism. Thus, it cannot be conclusively stated that the differentiated expression of estrogen receptors is a consequence of cryptorchidism, but rather a result of testicular injury due to the ectopia of the testicles.

Author response: Regarding the question on “ectopic testicle” raised by the reviewer, we would specify that cryptorchid testis examined do not show testicular degeneration (if the meaning refers to neoplastic degeneration). The cryptorchid condition is characterized by a block of spermatogenesis. 

Lines 420-421: this is overstated because you have not performed a histological analysis of the ectopic testicles in order to affirm an ongoing neoplastic transformation.

Author response: We thank the reviewer for his/her careful comment and eliminated the sentence.  

Lines 433-437: Why have you neglected your results of western blot SOD and Nfr2? Try to make a more tightened connection between your results and the literature ones.

Author response: We thank the reviewer for his/her suggestion. Regarding this point, we added some sentences to better describe the results of western blot relative to SOD and Nfr2 in the context of the literature as reported: “Our western blot results relative to SOD and Nfr2 are consistent, with that demonstrated by Zhao et al. [Experimental Gerontology, (2021), 152, 111460] in a study performed on rat testis during aging, where a decreased activity of SOD and of Nrf2 expression levels were associated to a reduced of seminiferous tubule diameters and of the seminiferous epithelium heights. Albeit the investigation was conducted in a different condition, the authors conclude that the observed downregulation of antioxidant ability mediated by Nrf2 pathway could be correlated with accumulation of oxidative stress increased DNA damage could be one of the causes implicated in the decline of testicular function during aging. Furthermore, a previous study by Li et al. [Reproductive Biology and Endocrinology 2013, 11:23] performed on mouse testis, demonstrated high levels of testicular and epididymal lipid peroxidation and associated low levels of antioxidants in the Nrf2 knock-out mouse” (see lines 495-504).

Conclusion

The conclusion is not aligned with the study's objectives. Try to adhere more directly to your data and refrain from making statements that were not directly observed in the present study. The conclusion needs to be rewritten.

Author response: We thank the reviewer for his/her comment that help us to improve the quality of manuscript. According to his/her suggestion, we changed the Conclusion (see the text “Our results describe for the first time the distribution and the expression of GPER in the testis-epididymal complex of cryptorchid dog. In particular, the increased expression of both GPER and ER-alpha along the tracts of cryptorchid epididymis is accompanied by a change in the morphology of the epithelium. These results represent the basis for further functional and molecular studies to better characterize the role of GPER in this physiopathological condition”) (see lines 552-555).

Round 2

Reviewer 1 Report

Comments and Suggestions for Authors

Although the paper has been improved and the authors have taken on board most of this referee's suggestions, there are still some critical issues that need to be resolved before it is accepted for publication.

Below I give suggestions for improving the quality of the paper.

Major concerns

 -  Line 254 and Table 2. The authors use the term 'gonocytes'. Since the gonocytes represent the fetal and neonatal stages preceding the formation of spermatogonial stem cells ( see Culty M, 2009) and normal testis contains spermatogonia, I suggest replacing “gonocytes” with “pre-meiotic cells”.

-          -  In Table 2 and Table 3 the authors speak of density as number of cells per unit area. Since they report neither the unit area considered nor the number of cells counted, it is likely that +++, ++, +,  +/- refer to the intensity of staining. If this is not the case, they should include  the method used for cell counting in the Materials and Methods section and edit the two tables.

 - In lines 279 and 291 “microvillus-covered surface” should be replaced with “stereocilia” because Schimming and Vicentini (2001) observed the stereocilia but not the microvilli in their ultrastructural study of the dog epididymis. The above correlates with scale bar values in Figure 2.

 Minor concerns

-         -  The title of Table 3 should be similar to that of Table 2, i.e. Immunohistochemical evaluation...

 -  The insets and letters in Figure 1 and 2 are not distinguishable.  It is therefore suggested to delimit them with a thicker frame and write the letters in a larger font.

 -    Please clarify what the type B cells mentioned in line 283 are.

Author Response

Although the paper has been improved and the authors have taken on board most of this referee's suggestions, there are still some critical issues that need to be resolved before it is accepted for publication.

Below I give suggestions for improving the quality of the paper.

 Author response: We thank the reviewer for the time spent for the revision process. We changed the parts of the manuscript according to the suggestions by the reviewer to improve the quality. 

 Major concerns

 -  Line 254 and Table 2. The authors use the term 'gonocytes'. Since the gonocytes represent the fetal and neonatal stages preceding the formation of spermatogonial stem cells (see Culty M, 2009) and normal testis contains spermatogonia, I suggest replacing “gonocytes” with “pre-meiotic cells”.

Author response: According to the suggestion by the reviewer, we changed the term 'gonocytes' with “pre-meiotic cells” in the text (see line 253) and in the Table 2.

  • In Table 2 and Table 3 the authors speak of density as number of cells per unit area. Since they report neither the unit area considered nor the number of cells counted, it is likely that +++, ++, +, +/- refer to the intensity of staining. If this is not the case, they should include the method used for cell counting in the Materials and Methods section and edit the two tables.

Author response: We thank the reviewer 1 for the comment and, as suggested, we modify the term “density” with “intensity of staining” in the description note of the Table 2 (see line 283) and Table 3 (see line 334).

In lines 279 and 291 “microvillus-covered surface” should be replaced with “stereocilia” because Schimming and Vicentini (2001) observed the stereocilia but not the microvilli in their ultrastructural study of the dog epididymis. The above correlates with scale bar values in Figure 2.

Author response: As suggested by the reviewer 1, we changed the term in “stereocilia” instead of “microvillus-covered surface” (see lines 289 and 304 Figure 2 legend).   

Minor concerns

-   The title of Table 3 should be similar to that of Table 2, i.e. Immunohistochemical evaluation...

The insets and letters in Figure 1 and 2 are not distinguishable.  It is therefore suggested to delimit them with a thicker frame and write the letters in a larger font.

Author response: As suggested by the reviewer, we changed the title of Table 3 by using the term “Immunostaining pattern” (see line 330) as just reported that of Table 2. Moreover, in order to make more distinguishable the insert and the letters inside them, as suggested by the reviewer 1, we increased the line width line and the font size for the letters.

 -    Please clarify what the type B cells mentioned in line 283 are.

Author response: As suggested by the reviewer, we modified the sentence as follows:” nonciliated type B cells containing vacuoles” (Ref. Wakui S, Furusato M, Takahashi H, Motoya M, Ushigome S. Lectin histochemical evaluation of glycoconjugates in dog efferent ductules. J Anat. 188:541-6) (see line 308).

Reviewer 3 Report

Comments and Suggestions for Authors

In general, the paper underwent significant changes, with the main focus now on the influence of the testicular ectopia on estrogen receptors in testis-epididymal complex.

I do not have any further comment and recommend publication the manuscript in the present form. 

Author Response

In general, the paper underwent significant changes, with the main focus now on the influence of the testicular ectopia on estrogen receptors in testis-epididymal complex. I do not have any further comment and recommend publication the manuscript in the present form. 

Author response: We thank the reviewer 3 for the time spent for the revision of our manuscript and for his/her appreciation of its improvement in term of quality thank to his/her important and careful suggestions.